# Participatory peer research exploring the experience of learning during Covid-19 for allied health and healthcare science students

Jean Daly Lynn[1]*, Lucia Ramsey[2], Joanne Marley[3], Johanna Rohde[4], Toni-Marie McGuigan[4], Adam Reaney[4], Brenda O'Neill[3], Andrea Jones[5], Danny Kerr[3], Ciara Hughes[6], Sonyia McFadden[7]

1 Lecturer in Psychology, School of Health Science, Ulster University, Ulster, Northern Ireland, 2 Lecturer in Occupational Therapy, School of Health Science, Ulster University, Ulster, Northern Ireland, 3 Lecturer in Physiotherapy, School of Health Science, Ulster University, Ulster, Northern Ireland, 4 Peer Researcher, School of Health Science, Ulster University, Ulster, Northern Ireland, 5 Lecturer in Podiatry, School of Health Science, Ulster University, Ulster, Northern Ireland, 6 Associate Research Director, School of Nursing and School of Health Science, Ulster University, Ulster, Northern Ireland, 7 Senior Lecturer in Diagnostic Radiography, School of Health Science, Ulster University, Ulster, Northern Ireland

* j.daly-lynn@ulster.ac.uk

**Data Availability Statement:** Data cannot be shared publicly because of ethical restrictions on sharing sensitive data. Data are available from the Ulster University Institutional Data Access (pure-

## Abstract

### Introduction

The teaching and learning experience of allied health and healthcare science students has altered because of the Covid-19 pandemic. Limited research has explored the experience on the future healthcare workforce using participatory research design. The aim of this study was to explore the impact of a global pandemic on the clinical and academic experiences of healthcare student using a co-production approach with student peer researchers.

### Methods

A participatory research approach adopting online focus groups facilitated by students trained as peer researchers was adopted. First, second and final year students from occupational therapy, physiotherapy, podiatry, healthcare science, diagnostic radiography and imaging, radiotherapy and oncology, and speech and language therapy were recruited to six focus groups. Data generated through focus groups were analysed thematically using the DEPICT model to support a partnership approach.

### Results

Twenty-three participants took part in six focus groups. The themes identified were: rapid changes to learning; living alongside Covid-19 and psychological impact. Students preferred blended learning approaches when available, as reduced peer interaction, studying and sleeping in the same space, and technology fatigue decreased motivation.

### Conclusion

Due to rapid changes in learning and the stress, anxiety and isolation created by the pandemic, managing study, personal life and placement resulted in a gap in confidence in

support@ulster.ac.uk) for researchers who meet the criteria for access to confidential data. This can be obtained through this link: https://pure.ulster.ac.uk/en/datasets/qualitative-participatory-peer-research-exploring-the-educational.

**Funding:** This work was supported by the Council for Allied Health Professionals Research (CAPHR) and The Centre for Health and Rehabilitation Technologies (CHArT). Both funds provided small amounts of money directly to the project team, CAPHR provided funds for interview transcription and CHArT provided funds for two peer researchers to engage in analysis. CAPHR: https://cahpr.csp.org.uk CHArT:https://www.ulster.ac.uk/research/topic/nursing-and-health/chart.

**Competing interests:** The authors report there are no competing interests to declare.

clinical skills development for students. Students took their professional identity seriously, engaged in behaviours to reduce transmission of Covid-19 and employed a range of coping strategies to protect wellbeing. A challenge with the move to online delivery was the absence of informal peer learning and students indicated that moving forward they would value a hybrid approach to delivery. Higher Education should capitalise on innovative learning experiences developed during the pandemic however it is important to research the impact this has on student skill acquisition and learning experience.

## Introduction

The Severe Acute Respiratory Syndrome-Coronavirus-2 also known as SARS-CoV-2 virus or Covid-19 was declared a pandemic in March 2020 by the World Health Organisation. At the end of 2021, over five million deaths worldwide had been attributed to Covid-19 [1]. Globally, it has been a public health crisis bringing significant societal changes and lasting consequences to health systems, education, and the economy. Rapid changes occurred in higher education to support continued delivery as education facilities in 194 countries closed [2]. There was a rapid move to online delivery as well as changes to delivery, reduced interaction, and access to physical resources [3]. The move from face-to-face delivery to online learning has been a challenging experience for everyone involved [4]. These challenges were further exacerbated by social inequalities and the digital divide [5]. Online learning is supported in the literature with adequate design and pedagogical basis, however the rapid migration meant that planning and design was not possible in response to the crisis [6]. Students as partners approaches, typically embedded in higher education to ensure a strong student voice in education delivery [7], were not prioritised during this transition [8]. The changes implemented due to Covid-19 need to be assessed and explored as it is unlikely that there will be a complete return to the traditional model of face to face teaching [9].

Allied Health Professionals (AHPs) refers to a diverse group of skilled professionals within the field of health and social care. AHPs are regulated by the Health and Care Professions Council (HCPC) as well as their respective professional bodies, and work autonomously or as part of a multi-disciplinary team in order to assess, diagnose, treat, rehabilitate and refer service users [10]. AHP disciplines include diagnostic radiography and imaging, occupational therapy, physiotherapy, podiatry, radiotherapy and oncology, and speech and language therapy. In addition, healthcare science is the study of diagnostic, analytic and monitoring procedures to evaluate the functioning of body systems. As part of student training, and in order to be eligible for HCPC and professional body registration, students must complete a minimum of 1000 clinical hours alongside theoretical studies [11].

Allied health education is considered highly complex in higher education [12]. As well as theoretical knowledge, healthcare students are required to learn skill and practice based content to augment their learning in preparation for clinical practice [13]. The change to the education format was particularly impactful for AHP students, due to the difficulty in learning clinical skills online [14]. Research has indicated that there is some merit for students in limited clinical skills teaching delivered online, however relying solely on online learning appears to pose challenges for both educators and students [15]. The practical nature of the curriculum in order for graduates to provide person centred care to patients can be a barrier to online delivery [16]. The impact of the rapid changes in modes of delivery are currently unknown [17].

Previous research has indicated that the pandemic has impacted the well-being and mental health of students [18]. Research studies have reported that at the start of the pandemic students were reporting high levels of stress [19,20]. In addition, research has indicated high levels of stress, burnout and psychological distress in frontline healthcare workers as a result of working during Covid-19 [16]. Less is known about the enduring impact of the pandemic particularly in allied health and healthcare science students with a curriculum centred on practical and clinical skills development. Students have reported feeling fearful of the possible future impact as they transition into the workforce [15]. Allied health and healthcare science students that have moved into the workforce earlier in response to Covid-19 have expressed concerns about their competencies for clinical practice [21]. However, it is not uncommon for students to feel unprepared as they transition to a new graduate role [22] irrespective of when training has occurred.

There is a growing recognition of the experiential knowledge held by students in relation to their learning and the expectation that students can be partners in education [23]. Student partnership is a process of equal involvement between educators and learners whereby students become active agents and share responsibility in common teaching and learning goals [24]. It shares attributes with participatory design such as collaboration, empowerment, autonomy and capacity building [25]. The rapid changes during the pandemic meant that students often lost their voice and control over their educational experience [8]. It is unlikely that healthcare education will fully return to the pre-pandemic modes of delivery, it is essential that the experiences of health professional students training during the Covid-19 pandemic are understood through the voice of the students concerned. This study was designed, conducted, and disseminated with students as partners to explore the impact on their learning experience and progression into the healthcare workforce.

## Aim

To explore the impact of a global pandemic on the clinical and academic experiences of allied health and healthcare science students.

## Methods

A participatory research approach with online focus groups facilitated by peer researchers was adopted to explore the experiences of allied health and healthcare science student experience during Covid-19. Students were involved from the development of the research protocol, data collection, data analysis, writing up the results and in dissemination activities. The DEPICT model [26] was used to underpin the thematic data analysis to provide a theoretical basis to the partnership approach. Ethical approval to conduct this study was granted by Ulster University Nursing and Health Research Ethics Filter Committee (FCNUR-20-020). The COREQ checklist was adopted to provide guidance in the reporting of this study [27].

### Underpinning theoretical approach

Participatory research incorporates methods that foster working 'with' rather than 'for' the community of enquiry [28]. Equality and partnership working are at the heart of this theoretical approach [29]. Bourke's [29] definition of participatory research involving the population being researched in the planning stages of the project right through to dissemination was adopted by the research team. The student voice was historically missing in educational research [30] and the role of student partners as co-inquirers is often ambiguous [31]. The participatory approach places the insider at the centre of the research process [32] and actively acknowledges this cohort as the expert in their experience [33]. The Covid-19 crisis resulted in

a loss of the student voice and limited partnership working due to the speed of changes [8]. The project team valued that student partners have a depth of insider knowledge through their lived experience, and therefore used participatory research as a conduit to co-construct knowledge. Peer researcher methodology was an approach adopted that recognised the insider knowledge of students and enables members of the student population to be agents and collaborators within the research team. It is suggested that data collection can be more authentic and valid when it is co-constructed with people that have lived through or with the experience [34]. This approach has been used within a wide range of populations, including older people [35], young people [36], people with long term health conditions [37] and minority communities [38]. The research team was conscious of hierarchy and power imbalances noted in the literature [39], and adopted strategies, such as regularly meetings and technologies that enabled duel control of inputting content, for all members of the project team to feel comfortable contributing on an equal level.

## Peer researcher approach

Six final year students from a range of programmes within the School of Health Sciences at Ulster University were recruited as peer researchers. One peer researcher was recruited into the project at the planning stage and informed the design of the project. After initial attempts to recruit via email across the entire final year student cohort were unsuccessful, students with notable communication skills were contacted directly by members of the project team. Emails were sent out detailing the requirements of the role (communication skills, organisation skills, digital literacy), time commitments and training expectations. A member of the research team (JDL) held individual and group meetings to discuss the role of the peer researcher. Interested students were sent the research protocol to read. Nine individuals explored the peer researcher role and six were selected on a first to volunteer basis. The six peer researchers were final year students from Podiatry (N = 1), Physiotherapy (N = 2), Radiotherapy and Oncology (N = 1), Occupational Therapy (N = 1), and Healthcare Science (N = 1). Typically, peer researchers have no prior research training. In this project, peer researchers had completed one research module and were in the process of completing a second level 6 module. The students were trained in qualitative research skills and ethical issues in research. The development of peer researcher training was guided by research, particularly the blended learning approach set out by Eaton *et al.*, [40]. Synchronous and asynchronous content was developed. A core focus of the training was on practicing communication skills and managing group discussions. The topic guides were co-produced from an initial draft document. The skills developed were then consolidated through each pair completing a pilot focus group.

Student peer researchers worked in pairs and had a clear protocol in the event of distress. Each pair had an academic member of staff waiting offline to provide formal or informal support. In addition, there was a thorough debrief after each focus group to ensure students had the opportunity to talk through and process the conversations that were discussed. These measures were important to ensure the psychological safety of all participants and researchers [41]. Confidentiality was an important component of the focus groups as peer researchers needed to maintain this outside of the project. It has been indicated that data collection can be richer as a result of peer researchers [42]. It is possible that students would feel more comfortable disclosing their experiences to their peers as opposed to their teachers in this project. Research has found that data in terms of questions asked and approach for peer researchers and academic researchers differ, however there is no evidence to indicate this impacts on data quality [43].

## Participant recruitment

Convenience sampling was undertaken within the student population registered for a programme within the School of Health Sciences. Students were in first, second and final year of the following programmes: Diagnostic Radiography and Imaging, Healthcare Science, Occupational Therapy, Physiotherapy, Podiatry, Radiotherapy and Oncology and Speech and Language Therapy. The focus groups were inter-professional within the same year. Students were invited to participate via email advertising. The email included a participant information sheet and a consent form. High attrition rates have been noted in computer mediated focus groups [44].

## Data collection

The focus groups were conducted using an online video conferencing platform called Blackboard Collaborate Ultra, that is part of the University Blackboard Learn platform. All students regularly use this platform to participate in teaching and learning. Once students had indicated their consent by returning their signed consent form, they were sent a weblink to the meeting time for their focus group. The weblink allowed the participant direct access to the online meeting space. At the start of the focus groups the academic member of staff introduced the peer researchers, outlined the supports in place in the event a student felt distressed and sought permission to record the focus group. The two peer researchers then assumed their roles as facilitators and the academic member of staff left the focus group. The peer researchers based their questioning on the topic guide. Following the focus group, the academic member of staff joined the collaborate session and debriefed the peer researchers. Focus groups were conducted during the 3rd UK national lockdown in February and March 2021.

## Data analysis

All members of the data analysis team undertook training on thematic analysis. Resources from Braun and Clark, the developers of reflective thematic analysis, were used to provide a foundation for training through YouTube lectures, academic papers [45–48], and their website (https://www.thematicanalysis.net). This provided appropriate baseline knowledge on coding and theme construction. An inductive approach was taken to the coding, with some semantic and latent codes developing as the team progressed. It was not possible to authentically follow this approach while collaborating with such a big team. A major deviation from Braun and Clarke approach was the use of a coding booking [48]. The aim was for the coding book to provide a collaborative support to coding with multiple coders, rather than a prescriptive approach restricting reflectivity and discussions were had regularly around meaning, streamlining terms used and introducing new codes. The coding for the team started independently and organically and evolved through researchers understanding of the data and subsequent collaboration. Reliability of coding was not measured, but discussed and shared interpretation provided the foundation for creating deep meaning within the data. This thematic analysis approach provided a flexible scaffolding to support the adoption of a framework [46]. The DEPICT model for participatory qualitative data analysis was adopted to provide a systematic framework [26].

The DEPICT model provided a framework for the analysis to ensure specific analytic steps. Table 1 presents a detailed description of the data analysis approach underpinned by the DEPICT model for participatory qualitative data analysis [26]. Regular team meetings with the student researchers and the coordinators ensured there was regular articulation of the analysis process and of the findings. The benefits of the diverse analysis teams enabled different viewpoints on the data and deeper interpretation. Rich and thoughtful discussion were the

**Table 1. Data analysis using the DEPICT framework.**

| DEPICT Steps | Team Responsible | Overview of Task |
|---|---|---|
| **Dynamic reading** | Team 1 Team 2 | Both team 1 and team 2 dynamically read the transcripts. The number of transcripts allocated per person were: Team 1: JDL (6); JR (4); RA (2) Team 2: JDL (6); AR (3); TMG (3) Team 1 watched the video recording and checked for accuracy and anonymity during the first read of the transcript. Each person was asked to keep a data analysis journal and to open the research questions on a separate word document. |
| **Engaged Codebook development** | Team 1 | A template was developed in Microsoft word to ensure the format of coding was undertaken the same was by the team. Two transcripts were independently coded (JR and RA) and each transcript was reviewed at a team meeting through screen sharing. Dynamic discussions were part of this process. The initial coding book was developed collaboratively identifying similar categories, generating labels, and documenting definitions for each label. The codebook was then piloted. The coding book was developed iteratively until a consensus of meaning was reached. When new codes were identified they were discussed in the team meeting. Regular meetings and review supported refinement of the codebook and identification of important ideas. |
| **Participatory coding** | Team1 | All transcripts were coded in Microsoft word. Following the development of the codebook the first transcripts were coded again to ensure accuracy. Each transcript was coded by one person and reviewed by a second person to establish consistency and generate deeper understanding. Transcripts were allocated as follows: JDL coded one transcript, reviewed three JR coded three transcripts, reviewed two RA coded two transcripts, reviewed one Each coded transcript was reviewed during a virtual meeting and independently to ensure consistency. Two students met independently to check meaning and interpretation at least once per transcript. Team meeting was held every two to three days over 6 weeks and individual supervision weekly with JDL. All peer researchers reviewed one transcript each after coding and their evaluation was fed into the analysis. |
| **Inclusive reviewing and summarizing of categories** | Team 2 | AR and TMG extracted the raw coded data into a word document aligned to the allocated categories for three transcriptions each. A shared folder managed version control and facilitated collaboration. A team meeting enabled the opportunity to reflect on important ideas and explore the data set in its entirety. The peer researchers key points for each transcript was discussed in relation to the data set. At this point all three team members had familiarity with the core data and categories. |
| **Collaborative analysing** | Team 2 | The Mural online platform was adopted to create a graphic representation of the key findings. Over two meetings (9 hours) categories were groups into potential themes using the bubble function. The coded extracts in a word document were continually consulted to keep grounded in the data. The online platform enabled the three people constructing the themes to move and alter the emerging thematic bubbles simultaneously. This empowered the student researchers to try out their interpretations as the analysis was developed. Rich discussions emerged from the different viewpoints. Three themes were developed as a result. |
| **Translating** | Team 2 | Each team member was allocated a theme and was responsible for writing up the meaning of the theme with corresponding quotations. JDL reviewed the findings to ensure it emphasised the important viewpoints throughout the data analysis process. These finding are presented in the results section. |

cornerstone of this collaborative approach. This sense making of data and exploration of meaning was adopted a way of validation in line with scholars in qualitative inquiry [49].

All focus groups were transcribed verbatim by a professional transcriber. The data analysis was undertaken in two separate parts with two different teams. All transcripts were coded by one person and reviewed or explored for any differences or further meaning by two people. All peer researchers reviewed one transcript of a focus group they undertook to ensure the correct meaning was interpreted. Each peer researcher was asked to provide their opinion of the three most important topics they read in the transcript. No major differences in meaning were identified between coding and the review process. This enabled a process of peer validation, while managing the peer researchers time commitment to the project, and opportunities to reflect on the findings as a team. Rigour was established through reflective journaling, regular meetings and partnership working, and detailing the data analysis approach in Table 1.

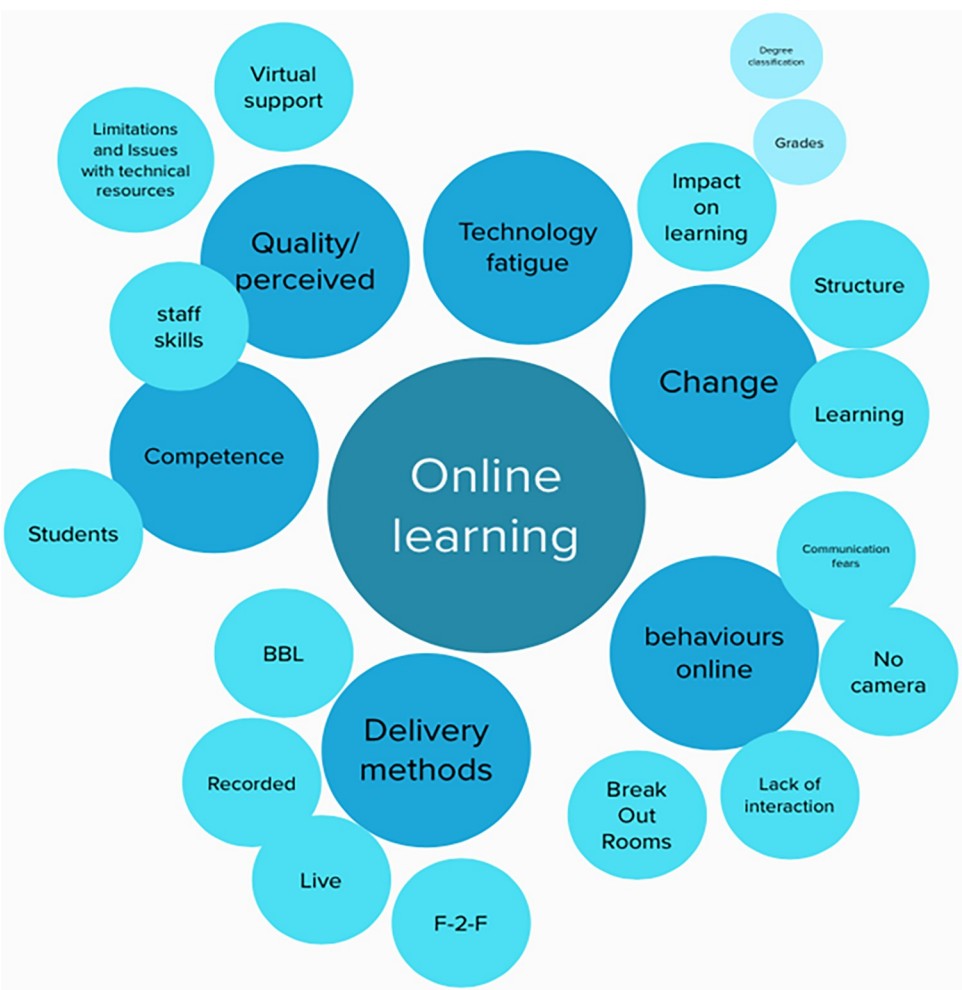

**Fig 1. An example (screenshot) of an early stage of co-developing themes using the platform Mural.**

Additionally, the consistency of one researcher (JDL) through data collection and analysis supported the rigorous analytic approach with multiple partners.

Mural (https://www.mural.co) was a platform that enabled simultaneous interaction and manipulation of content. Codes were inputted into the platform, during discussion and exploration of the data set grouping of the codes according to patterns that developed were tested out visually. Fig 1 represents a sub-set of the data testing out an initial pattern. Discussion over nine hours, with continued identification of patterns and independent reflection, resulted in the final thematic representation illustrated in Fig 2.

## Results

Twenty-three participants participated in six focus groups ranging from 2 to 5 participants. A total of six first year, nine second year and eight final year students participated. Participants were studying Occupational Therapy (N = 8), Physiotherapy (N = 10), Healthcare Science (N = 2), Diagnostic Radiography and Imaging (N = 1) and Speech and Language Therapy (N = 2). Seven participants were male (16 female) and eleven were mature students, defined as over twenty-one years of age. Focus groups were on average 74.5 minutes long, ranging from 39 minutes to 99 minutes. No technical issues impacted on participation were reported. The

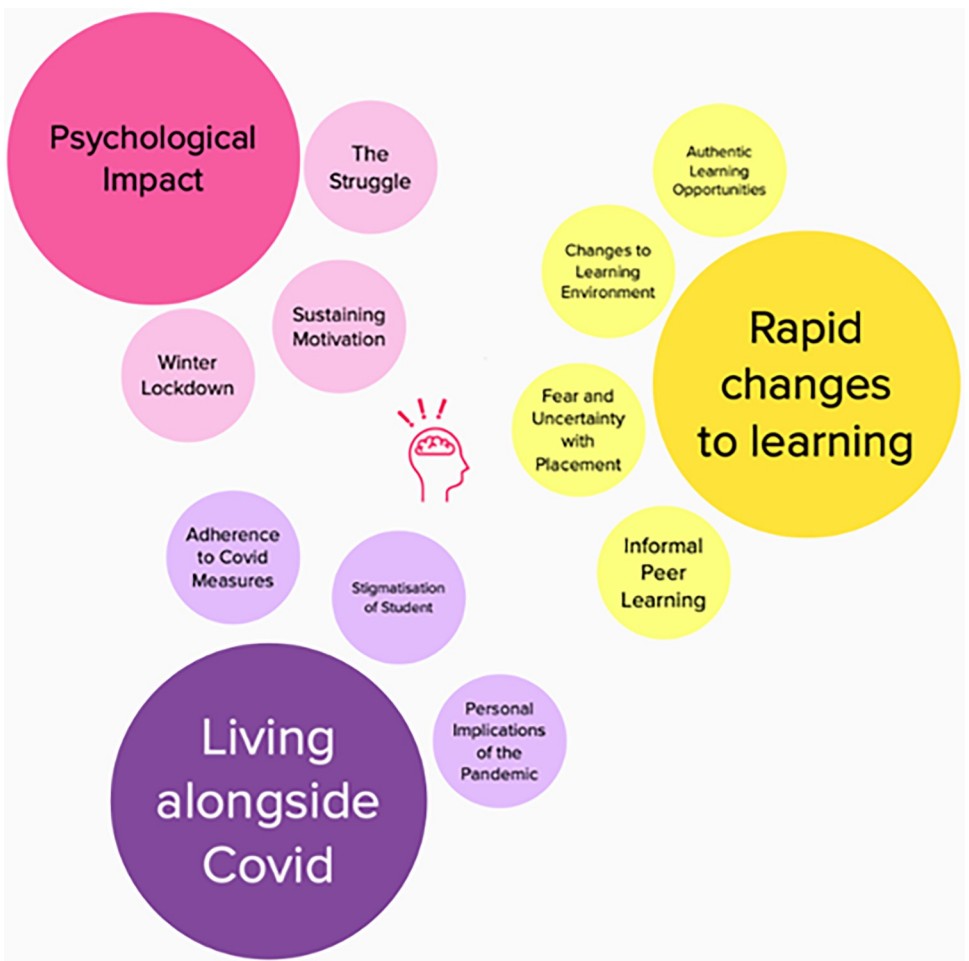

**Fig 2. Extract of final themes co-developed by student and academic researchers on the platform mural.**

three themes developed from the thematic analysis (Fig 2) were: *rapid changes to learning*; *living alongside Covid-19* and *psychological impact*. Pseudonyms were given to participants.

## Rapid changes to learning

The educational impact from the rapid changes to learning was a significant theme due to the COVID-19 pandemic. The four sub-themes were: *Authentic Learning Opportunities, Impact on Learning, Fear and Uncertainty about Placement* and the *Informal Peer Learning*

**Authentic learning opportunities.** It was evident that learning and studying from home had a huge impact on students. Participants felt the lack of authentic learning experiences, they described as hands-on experience and in-person classes, adversely impacted their learning. There was a sense of concern that this would impact assessment results, '*I think it will impact my grades as well, my final grade, because of what we had to go through*' (Noah, Year 3 [Y3], Focus Group [FG]2). Many participants felt they were not given an authentic learning experience when practical classes were held online. The belief was that face-to-face practical skills classes were required, '*a lot of stuff is hands-on, and that aspect was kinda taken away from us. . .so when we went out on placement, it made it a lot harder, it was difficult to pick up on the smaller things*' (Marian, Y3, FG1). Participants felt the lack of practical classes resulted

in feeling less confident when interacting with service users on placement. Many believed that if they got to practice and use the required equipment on campus, their learning and skills would be greatly improved.

**Change to learning environment.** The environment in which students learned, altered significantly. Discussions included teaching space, study space, accommodation challenges, access to placement, campus visits to learn clinical skills and the impact personal circumstances had on learning because of Covid-19. Many students found the change of learning environment from university to home difficult. This was due to lack of study space, inadequate Wi-Fi and technical resources. Numerous participants reported the challenge of having several people in the household using technical resources. '*The internet connection wasn't good because there were five people using the internet for calls at the one time. So, it really wasn't ideal*' (Dara, Y2, FG2). Additionally, participants attended classes and took exams, in the place they also slept and ate. Participants stated that spending so much time in one room on the computer was difficult and tiring. Although participants experienced many challenges, some benefits of learning from home were the online and recorded lectures; '*Been great for recorded lectures that were done on Collaborate, being able to go back over them right before exams*' (Catherine, Y2, FG1).

**Fear and uncertainty about placement.** Although placement continued for students, it generated worry and anxiety. The nature of the pandemic meant that health and social care guidance changed frequently and rapidly. This resulted in uncertainty around dates and location of placement, as well as last minute changes. The uncertainty created feelings of frustration and for some participants, this meant uncertainty regarding accommodation, '. . .*because if we do go on placement, I'll need somewhere to stay*' (Chloe, Y1, FG1). All participants reported that '*the unknown,*' (Sarah, Y2, FG1) around placement was a consistent worry, particularly around opportunities to learn and develop skills when on placement.

Participants were worried about going into an environment where Covid-19 transmission risk increased and feeling unprepared in relation to skill development. '*I didn't feel I was getting the full experience of what I would have as a mental health OT. . .. To know that we are missing out on being able to get the full experience is disheartening*' (Tara, Y3, FG2) Additionally, concerns about lack of support were reported for example, if their practice educator had to isolate or was under significant pressures. '*It isn't ideal because the radiographers that are in the hospital don't really have the time to teach us a whole lot of stuff*' (Joe, Y2, FG2). The lack of classroom-based practice reduced confidence in clinical ability going into placement. One participant reported, '*I am worried going into final year. . .that some practice educators might expect you to know everything*' (Alison, Y2, FG1).

## Informal peer learning

The move to online learning where social interaction between peers dramatically decreased highlighted the importance of peer-to-peer learning. Information exchange of thought, questions, checking understanding and developing ideas together regarding lectures, assignments, or tasks was identified as a core element of learning that was significantly reduced. This opportunity negatively impacted student's motivation, and created unmet needs around encouragement, and reassurance, '*I wish I was at Uni, so that I knew where everybody else was at and maybe it would give me the motivation to get going*' (Tara, Y3, FG2). The lack of in-person informal peer learning hindered their learning development and confidence with many stating that they simply missed the reassurance of, '*bouncing ideas off for assignments and that kind of thing*' (Caitlyn, Y1, FG1).

One participant lived with classmates during the pandemic stating, '*I don't know how I would have passed an exam or done an assignment without them.*' (Jordan, Y3, FG2). The use of text messaging and social media groups were widely reported with mixed opinions on their usefulness. The lack of informal conversations between and during class also adversely impacted students, '*. . .you're missing out on that small chitchat, [social media] isn't the same. . .*' (Sharon, Y3, FG1). This was particularly challenging for first year students beginning their university journey '*I think the hardest part was the change from secondary school to university and then not knowing people on my course, not being able to meet them, so you can't ask for help with stuff you are struggling with*' (Annmarie, Y1, FG1).

### Living alongside Covid-19

Three sub themes were identified under the significant theme of Living alongside Covid-19: *Adherence to Covid-19 Measures, Stigmatisation of Students and Personal Implications of the Pandemic.*

**Adherence to Covid-19 measures.** It was evident that the participants adopted professional behaviours to minimise Covid-19 transmission. For example, '*wash your hands regularly during the day when you're on placement and then coming home, scrubbing yourself head to toe and washing your clothes regularly and stuff like that*' (Jordan, Y3, FG2). Although there was some evidence of students travelling from student accommodation to their family home during restricted periods it was apparent that participants made attempts to reduce virus transmission. Participants reported feeling safe on campus when learning clinical skills and valued the measures put in place to reduce the risk of catching Covid-19; '*There are so many measures that are put in place to try and prevent the spread of Covid, obviously, it's not the same, it's a bit different with all the PPE and they always have to keep the windows open, so the classrooms are freezing*' (Uma, Y1, FG2). Despite the rigorous efforts made by students to reduce Covid-19 transmission some students did catch Covid-19 themselves noting that it did have a psychological effect on them; '*My entire family tested positive actually. When you get it, it hits you psychologically, actually*' (Noah, Y3, FG2).

**Stigmatisation of students.** It was clear that the students had a fear of passing Covid-19 to vulnerable people; '*I think it is maybe not so much for me catching it. My mum is high risk and then obviously I have a young family*' (Ellen, Y3, FG2). Participants reported leaving their part-time jobs so that they could reduce the risk of transmission to vulnerable service users during their placement. Additionally, participants sought alterative accommodation for placement, so they were not putting their family at risk. A poignant piece to note from the focus group was that the students felt the media portrayed them in a negative light. Participants reported diligently following guidance and were disappointed about attitudes towards students on placement because of media coverage; '*I do think at times there was this blanket narrative that the students were the ones, and they were going out wild a couple of months ago and there was a huge problem. . .but those were the few compared to a lot of students who really were sticking to the regulations, the majority of students that actually are at home and are not on campus*' (Chloe, Y1, FG1).

**Personal implications of the pandemic.** There were a range of both positive and negative implications for participants because of Covid-19 related restrictions. Participants appreciated the reduced distractions which allowed them to focus on their work; '*I feel like it has helped me to get things done on time because I have nothing else to concentrate on*' (Sharon, Y3, FG1). One participant stated that they do not know how it would be possible to undertake the workload of final year if they were socialising and engaging in leisure activities. Other benefits included feeling more supported by lecturers and spending more time with family.

Some challenges included frustrations felt about student accommodation. Participants reported signing contracts for accommodation before the University announcement of a move to online learning and subsequently paid for unnecessary accommodation; '*I have signed a contract as well, but I haven't been back since before Christmas. I will still be paying for it until the end of the year, and I don't think I will be back*' (Marion, Y3, FG2). Equally, students living with at-risk people had to stay outside of the home when attending placement to reduce the risk to vulnerable people resulting in financial implications for participants.

Furthermore, students who are clinically high risk could not be physically present on campus or attend placement due to Covid-19 risk; '*I have Crohn's Disease. This means I'm in the high-risk category for Covid, so the Trust weren't willing to take the risk of me coming out onto the wards, even with all the PPE. So, that will be two placements, now, I have to make up within the next year-and-a-half*' (Paul, Y2, FG2). This has a long-term impact on their ability to meet the learning requirements of their programme.

## Psychological impact

Three sub themes were identified under Psychological Impact: *The Struggle, Winter Lockdown and Sustaining Motivation*.

**The struggle.**   There was a real sense that participants struggled during Covid-19, managing workloads, responsibilities, part time jobs, risk and caring for others. For example, 'the *struggle for me has been juggling everything*' (Annmarie, Y1, FG1). Participants wanted empathy for their personal circumstances, which ranged considerably, and a number of participants had caring responsibility for young children (Haley, Dorian, Tara, Paul, Sarah, Ellen). '*Trying to keep three children socialised and occupied, while being a student and trying not to let it have to be an impact*' (Haley, Y1, FG1). Participants felt there was little practical support available during this time for people. The focus groups highlighted the importance of feeling more supported by the university through informal communication by lecturers to check on student welfare and mental health and wellbeing, '*A wee bit of a, "hey everyone, how's it going? Things have been . . ." not just an email saying, "anybody that's in need of support, here's links." I think maybe, from a mental health perspective, there could have been a wee bit more support*' (Ellen, Y3, FG2).

The lack of contact with other students meant participants often did not know how other people were feeling and if they were having similar struggles. Again, this highlighted the isolating and lonely experience students had during their studies. Feeling more supported during this time could have helped the learning capacity of students. This was also evident when participants unanimously reported the challenges with reduced social interaction because of Covid-19 restrictions; '*the biggest struggle has probably been the social side of lockdown; not being able to see my friends that I would see from around home, and then not being able to get into Uni as well, was really hard*' (Joe, Y2, FG2).

Zoom calls were beneficial for keeping in touch with friends and family however, as time progressed, technology fatigue set in, and the lack of social contact created isolation. WhatsApp and Facebook groups supported the maintenance and building of relationships though some participants reported leaving group chats. There were differences between participants from different courses as some programmes were fully online whereas others attended on campus practical's every three weeks. This impacted the ability to make and build friendships. Year 1 participants agreed that one advantage of working in smaller bubbles was that it facilitated deeper friendship; '*It's actually been great because you spend more times with these ten people and you get to know them better that if you were in a big group of 80 or 90*' (Annmarie, Y1, FG1).

**Winter lockdown.** It was evident that there was a huge psychological impact during the third period of Covid-19 restrictions. At the time of data collection participants had lived through nearly a year of Covid-19 related restrictions. Participants felt more negative feelings during the winter period as Zoom calls stopped, the 'new' aspect of lockdown had worn off and isolation from peers was experienced; '*Definitely, in the middle of lockdown you did feel isolated because you weren't seeing your mates. . .you did really feel isolated from your surrounding friends and other family*' (Tara, Y3, FG1). Additionally, participants felt they had to be cautious of Covid-19 all the time. The extended screen time impacted on psychological wellbeing; '*It's hard to get away from the screen, it sometimes gets you down*' (Haley, Y1, F1).

Resilience was also evident with participants reporting they continue to work hard through the challenges. A range of coping strategies were identified within the focus groups. These included taking breaks, exercise, visualisation, adult colouring books, meditation, routine, organisation and chatting to a friend. A common theme was the importance of going outside for fresh air and light particularly in the winter; '*the simple importance of getting up and getting out into the daylight and going for a walk*' (Dorian, Y1, F1).

Organised sporting activities and hobbies were missed, and participants reported taking up new activities such as running; '*I have taken up running, which was again, to get me out of the house because all my hobbies have shut down with Covid*' (Catherine, Yr2, FG1).

**Sustaining motivation.** The online lecture delivery impacted participants motivation. For some participants, they felt all day was spent at their computer listening to lectures. For other participants, they had limited live teaching experiences and were expected to work independently through content. The level of independent work was reported as difficult to sustain motivation. Participants reported having the time to engage but not the impetus to. Additionally, there were a range of distractions in the home *environment; 'I have definitely had to try and put a routine in place because if I don't, I would literally just watch Netflix all day*' (Damien, Y2, FG2).

Studying in the same environment you are sleeping was demotivating and impacted participants ability to sleep. Additionally, the lack of physically attending campus was reported to decrease motivation; '*because you were sitting at home, you just couldn't shift your focus to learning, whereas whenever you get in the car, I travel over an hour to get to Uni, whenever you get in the car, you're preparing yourself to go to learn and you're switching off from home, but that two-minute walk from the bedroom to the kitchen table isn't the same*' (Tara, Y3, FG2).

## Discussion

The aim of the present study was to explore allied health and healthcare science student experience of learning during the Covid-19 pandemic through student researchers. A novel aspect of this study was that peer researchers used their lived experience to facilitate the focus groups and analyse the findings. The DEPICT model was instrumental in the partnership approach to data analysis. The findings indicated that students preferred blended learning approaches, while ensuring there were adequate opportunities for informal peer learning. Additionally, the healthcare workforce of the future will need further support. The well-being of these emerging professionals is of paramount importance considering that their skills and identities developed during a turbulent time of rapid change, isolation, and heightened anxiety.

While clinical placement was undertaken during this period, an element of uncertain and last-minute changes was apparent. This differed from international experiences where clinical placement was postponed [10] or replaced by telehealth approaches and simulated exercises [50]. The delivery of placement was challenging on the part of those involved in its organisation and facilitation, and generated worry and anxiety with students. The findings indicated

that students found the constant struggle of placement and educational workloads, while trying to stay safe and caring for others particularly challenging. This had a psychological impact of anxiety, stress, low motivation and decreased social interaction that was reflected in previous research [3,50–52]. This research took place during the third lockdown in the United Kingdom situated in the winter months (February-March 2021) possibly adding to the demotivated feelings reported. The wellbeing and experiences of students is paramount to their learning. Further exploration into the way students interact with professional services such as librarians and student support in an online environment could highlight ways to enhance this service provision in the future [53].

The findings from this work reflect other research that has indicated allied health students had high levels of positive attitudes and health behaviours towards the pandemic [54], although they were worried about spread of infection to themselves and others. Findings demonstrated the personal sacrifices students made to protect vulnerable family and service users even when impacted by negative financial implications. The pandemic is still prevalent, indicating the need for further consideration of ways to support students with secondary conditions to safely meet their required placement hours and their learning needs. For example, within this study, research placements were utilised to support students to build up their clinical hours where professional competencies had previously been met in a clinical setting. In line with previous research, students were fearful they would not accomplish their professional skills and competencies through an online teaching environment [51]. The long-term impact on the future workforce is currently unknown [17]. Future preparation for practice may be needed to ensure confidence in skill level is increased and development is supported, as students across three years of a programme impacted by Covid-19 emerge into the workforce.

Research has indicated that online learning methods are as effective as face-to-face approaches to achieve learning outcomes [55]. Online learning is not recommended for this student cohort moving forward, blended learning is a preferred approach, and it is important to capitalise on the learning from this experience and consider the long-term impact of changes made quickly in response to a crisis [56], ensuring that pedagogical rationales clearly underpin decision making and student partnership models are at the heart of this process [23]. The expectation of digitalisation in education is likely to have a lasting impact. For healthcare students, an enhanced blended learning approach should be optimised [57]. A holistic framework for enhanced blended learning would enable graduates to harness the necessary skills to work in healthcare as the digital landscape changes while also mastering their clinical competencies in a classroom setting. The findings support the future direction of learning needs as we move into a post-Covid-19 era.

In line with previous findings [15], students' inability to learn from their peers had a negative impact on their knowledge acquisition. Future approaches need to consider the social process of learning to maximise the learning capacity of students in the classroom and online environment. Social learning theory postulates that collaborative working, and shared learning experiences enhance learning outcomes [58]. Rich interaction is key to online teaching and learning approaches [59]. The social process of learning that is required to take place in a social context was impaired through the online environment [58]. The peer learning notable in the findings was also the informal occurrences that happened over coffee, through body language and between classes. Therefore, reports of feeling lonely and demotivated were evident.

A limitation of this study was that research was conducted in a single department in one UK based university. The experience for students studying similar courses in other universities could be different. For example, the authors are aware that not all universities were able to sustain delivery of clinical placements during the pandemic. Additionally, a low response rate was found, and this could have been because of the considerable pressures that students felt they

were under at this time. Future research should explore the needs of students going into the workforce to mitigate the impact of missed learning opportunities. Additionally, it would be interesting to explore the uptake of online mental health services given the degree of stress and anxiety reported by the student population. A future consideration should be given to the inclusion of students in research. Involvement in this study enabled students to gain first-hand research experience, have their voice heard and get an appetite for research that they can carry into their healthcare career.

## Conclusion

This study indicates that due to rapid changes in learning and the stress, anxiety and isolation created by the pandemic, managing study, personal life and placement resulted in a gap in confidence in clinical skills development for students. Despite this, the students took their professional identity seriously and engaged in behaviours to keep those around them safe from the transmission of Covid-19. A notable challenge with the move to online delivery was the absence of informal peer learning, and students indicated that moving forward they would value a hybrid approach to delivery. As such, Higher Education should capitalise on innovative learning experiences developed during the pandemic, including virtual and research placements, however it is important to research the impact this has on student skill acquisition and learning experience.

## Acknowledgments

Thank you to the wider team including academic researchers within the School of Health Sciences at Ulster University, peer researchers and placement student.

## Author Contributions

**Conceptualization:** Jean Daly Lynn, Lucia Ramsey, Joanne Marley, Johanna Rohde, Brenda O'Neill, Andrea Jones, Danny Kerr, Ciara Hughes, Sonyia McFadden.

**Data curation:** Jean Daly Lynn, Lucia Ramsey, Joanne Marley, Johanna Rohde, Toni-Marie McGuigan, Adam Reaney.

**Formal analysis:** Jean Daly Lynn, Lucia Ramsey, Joanne Marley, Johanna Rohde, Toni-Marie McGuigan, Adam Reaney.

**Funding acquisition:** Brenda O'Neill, Ciara Hughes, Sonyia McFadden.

**Investigation:** Jean Daly Lynn, Johanna Rohde, Toni-Marie McGuigan, Adam Reaney, Sonyia McFadden.

**Methodology:** Jean Daly Lynn, Lucia Ramsey, Brenda O'Neill, Andrea Jones, Danny Kerr, Ciara Hughes, Sonyia McFadden.

**Project administration:** Jean Daly Lynn, Ciara Hughes, Sonyia McFadden.

**Resources:** Ciara Hughes, Sonyia McFadden.

**Supervision:** Jean Daly Lynn.

**Validation:** Jean Daly Lynn, Ciara Hughes, Sonyia McFadden.

**Writing – original draft:** Jean Daly Lynn, Lucia Ramsey, Toni-Marie McGuigan, Adam Reaney, Ciara Hughes, Sonyia McFadden.

**Writing – review & editing:** Jean Daly Lynn, Lucia Ramsey, Joanne Marley, Johanna Rohde, Toni-Marie McGuigan, Adam Reaney, Brenda O'Neill, Andrea Jones, Danny Kerr, Sonyia McFadden.

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
