## [Decision Letter · Decision Letter 0]

1 May 2022

PONE-D-22-09516Qualitative participatory peer research exploring the educational impact of Covid-19 on Allied Health and Healthcare Science StudentsPLOS ONE

Dear Dr. Daly-Lynn,

Thank you for submitting your manuscript to PLOS ONE. After careful consideration, we feel that it has merit but does not fully meet PLOS ONE’s publication criteria as it currently stands. Therefore, we invite you to submit a revised version of the manuscript that addresses the points raised during the review process.

We look forward to receiving your revised manuscript.

Kind regards,

Gwo-Jen Hwang

Academic Editor

PLOS ONE

Journal Requirements:

"This work was supported by the Council for Allied Health Professionals Research (CAPHR) and The Centre for Health and Rehabilitation Technologies (CHArT). "

"This work was supported by the Council for Allied Health Professionals Research (CAPHR) and The Centre for Health and Rehabilitation Technologies (CHArT). 

Both funds provided small amounts of money directly to the project team, CAPHR provided funds for interview transcription and CHArT provided funds for two peer researchers to engage in analysis. 

CAPHR: https://cahpr.csp.org.uk

CHArT:" ext-link-type="uri" xlink:type="simple">https://www.ulster.ac.uk/research/topic/nursing-and-health/chart"

Reviewers' comments:

Reviewer's Responses to Questions

**Comments to the Author**

1. Is the manuscript technically sound, and do the data support the conclusions?

Reviewer #1: No

Reviewer #2: Partly

2. Has the statistical analysis been performed appropriately and rigorously? 

Reviewer #1: No

Reviewer #2: N/A

3. Have the authors made all data underlying the findings in their manuscript fully available?

Reviewer #1: No

Reviewer #2: Yes

4. Is the manuscript presented in an intelligible fashion and written in standard English?

Reviewer #1: No

Reviewer #2: Yes

5. Review Comments to the Author

Reviewer #1: Dear Authors,

The study tried to use the DEPICT Framework to Analysis the interview result is good, yet, suggest authors address more related interview guidelines, coding book, coding analysis method, for instance, how to get the Themes and sub-themes by table 2, the section unclear.

On page 7, "…we aimed to recruit 10 participants to each session….” But the Results section shows that “Twenty-three participants participated in six focus groups ranging from 2 to 5 participants….” So what are the actual samples?

-On page 7, Although the authors presented the “Table 1: Data Analysis using the DEPICT Framework,” what’s the detailed codebook related Themes and sub-themes also unclear, so how to mentation the result is contributed.

-On page 7, “The focus groups were conducted using the platform Blackboard Collaborate Ultra” the platform should address how to recruit different professional field participants to join the platform.

- Data analysis section, “The data analysis was undertaken in two separate parts with two different teams. All transcripts were coded by one person and reviewed by two people.” For more contributions, please add Coder reliability and validity.

-up to the topic is “Qualitative participatory peer research exploring the educational impact of Covid-19 on Allied Health and Healthcare Science Students” however the study related to educational activities on Allied Health and Healthcare Science Students is unclear, so the topic may need modification.

Thank you.

Reviewer #2: The study has attempted to examine the impact of Covid-19 on health science students. This study was carried out using qualitative research methodologies. The instrument used in doing research was an online focus group. The focus group was facilitated by the peer researchers, so it is called qualitative peer research. It starts with the literature outlining the impacts of Covid-19 in overall sectors of life including education. The aim of the research is to explore the experience of allied health and healthcare students. The method adopted here was participatory action research (PAR) with online focus group facilitation. The findings of the study cover health and education-related issues. This study is appropriate to assemble the information regarding the impact of Covid-19 on health sector students. However, I have some remarks on the paper.

1.Title is too long and everything is included in the title: Qualitative participatory peer research…… I think, Participatory peer research exploring the educational….. is appropriate.

2.In the Abstract, some claiming sentences are written at the beginning of the abstract without any evidence and appropriate support. How could you say the global pandemic had an impact on teaching and learning activities in health and health care students?

3.On page 2, line 17, it is written that the knowledge generated from this project was co-production using an online format, including the dissemination. What does it mean?

4.In the introduction section, different previously done researches are mentioned. But, a single paragraph gives multiple ideas, which is not good in academic writing. In the introduction, a discussion about allied health students is missing. This could be new for non-health sector readers.

5.In the methods section, it is mentioned that participatory action research (PAR) was adopted in the research. But, this study does not carry any essence of PAR. Participatory action research requires participants to build and keep evidential records of practice, theory, and reflection and to provide a reasoned justification to others for their work. Doing this requires authentic participation and is collaborative, establishing self-critical, non-hierarchical communities and partnerships. It is also a recursive and systematic process of learning, with planning, action, analysis, and reflection leading to further planning, action, analysis, and reflection. Only, the peer research would be appropriate but it is not action research. The methods section also misses why peer research was used.

6.Ethical issues in doing the qualitative peer research are missing. How these issues were handled?

7.Conclusion of the study does not hold the essence of the overall result and discussion.

At last, this paper could be made publishable after the rigorous work in the methods section. Does it actually a PAR? Does it complete all the processes of PAR?

6. PLOS authors have the option to publish the peer review history of their article (what does this mean?). If published, this will include your full peer review and any attached files.

Reviewer #1: No

Reviewer #2: No

---

## [Author Response · Author response to Decision Letter 0]

14 Jul 2022

Thank you for the opportunity to enhance this manuscript. A response to the reviewers has been included in this resubmission with each comment addressed.

We believe the manuscript is formatted according to style requirements

Funding Statement:

This work was supported by the Council for Allied Health Professionals Research (CAPHR) and The Centre for Health and Rehabilitation Technologies (CHArT). 

Both funds provided small amounts of money directly to the project team, CAPHR provided funds for interview transcription and CHArT provided funds for two peer researchers to engage in analysis. 

CAPHR: https://cahpr.csp.org.uk

CHArT:https://www.ulster.ac.uk/research/topic/nursing-and-health/chart

Any funding related information was removed from the manuscript.

Data availability

Data cannot be shared publicly because of ethical restrictions on sharing sensitive data. Data are available from the Ulster University Institutional Data Access (pure-support@ulster.ac.uk) for researchers who meet the criteria for access to confidential data. This can be obtained through this link: https://pure.ulster.ac.uk/en/datasets/qualitative-participatory-peer-research-exploring-the-educational

Reviewer 1 comment Response to reviewer

The study tried to use the DEPICT Framework to Analysis the interview result is good, yet, suggest authors address more related interview guidelines, coding book, coding analysis method, for instance, how to get the Themes and sub-themes by table 2, the section unclear.

 Thank you for highlighting that the data analysis section was unclear. This section has been revised to take into consideration the core principles of thematic analysis that was used within the DEPICT framework. It is great to have to opportunity to expand on this and make it clearer.

For transparency, this approach was guided by the work of Braun and Clark (multiple publications although the reference below would be helpful to refer to for clarity). 

I am not completely sure what this reviewer is asking for as I believe we have experience of different qualitative approaches. I have attempted to generate more clarity in the development of codes into themes through the Mural images (Figure 1 and 2). 

Braun V, Clarke V. Can I use TA? Should I use TA? Should I not use TA? Comparing reflexive thematic analysis and other pattern-based qualitative analytic approaches. Couns Psychother Res. 2021;21(1):37–47.

On page 7, "…we aimed to recruit 10 participants to each session….” But the Results section shows that “Twenty-three participants participated in six focus groups ranging from 2 to 5 participants….” So what are the actual samples?

 This statement has been removed for clarity. The sample is 23 participants. 

-On page 7, Although the authors presented the “Table 1: Data Analysis using the DEPICT Framework,” what’s the detailed codebook related Themes and sub-themes also unclear, so how to mentation the result is contributed.

 To provide clarity around the themes and codes Figure 1 is a screenshot of early codes in a subset of the data. Figure 2 presents the final themes and sub-themes through the Mural platform. 

-On page 7, “The focus groups were conducted using the platform Blackboard Collaborate Ultra” the platform should address how to recruit different professional field participants to join the platform.

 I am unsure what this reviewer is asking for. If I have not addressed it properly, I am happy to with more clarity. 

The following was added: ‘The focus groups were conducted using Blackboard Collaborate Ultra, that is part of the University Blackboard Learn platform. All students regularly use this platform to participate in teaching and learning.’

- Data analysis section, “The data analysis was undertaken in two separate parts with two different teams. All transcripts were coded by one person and reviewed by two people.” For more contributions, please add Coder reliability and validity.

 Further clarity on the data analysis was expanded upon and approaches used to support the trustworthiness and credibility of the data was described. Rigor was achieved through reflexive journaling, peer researcher review, regular meetings, and deep discussion. Participatory data analysis uses the subjective interpretation of the data by people with lived experience. Through discussion and reflection of this experience richer data collection, discussion and interpretation could but achieved.

I am not sure if the reviewer is looking for inter-rater reliability. This was not in line with our approach. This does not support the rich multiple coder approach taken in the study. See this paper for supporting evidence. 

Sweeney, Angela, Kathryn E. Greenwood, Sally Williams, Til Wykes, and Diana S. Rose. "Hearing the voices of service user researchers in collaborative qualitative data analysis: the case for multiple coding." Health Expectations 16, no. 4 (2013): e89-e99.

-up to the topic is “Qualitative participatory peer research exploring the educational impact of Covid-19 on Allied Health and Healthcare Science Students” however the study related to educational activities on Allied Health and Healthcare Science Students is unclear, so the topic may need modification.

 The manuscript refers to the process (peer researchers) and outcome (experience of covid). It is difficult to unpick the two as peer researchers explored the impact Covid had on their peers, and they are all allied health and healthcare science students. I hope that the revisions within the manuscript help to clarify and mesh the process and outcome into a seamless thread throughout the paper. 

Reviewer 2 comment 

1.Title is too long and everything is included in the title: Qualitative participatory peer research…… I think, Participatory peer research exploring the educational….. is appropriate.

 Thank you for this recommendation. The word Qualitative has been removed from the title in line with this recommendation. 

2.In the Abstract, some claiming sentences are written at the beginning of the abstract without any evidence and appropriate support. How could you say the global pandemic had an impact on teaching and learning activities in health and health care students?

 This sentence was changed to say that the teaching and learning has altered for students, as all campus-based teaching and learning activities did during the pandemic. As it is not common practice to use citations in an abstract, I hope this is an appropriate change in line with your recommendation. 

3.On page 2, line 17, it is written that the knowledge generated from this project was co-production using an online format, including the dissemination. What does it mean?

 In line with the revision made to the conclusion this sentence was removed. 

4.In the introduction section, different previously done researches are mentioned. But, a single paragraph gives multiple ideas, which is not good in academic writing. In the introduction, a discussion about allied health students is missing. This could be new for non-health sector readers

 The introduction section has been revised and I hope this makes it easier to read with single ideas presented per paragraph. An introduction to allied health professions and healthcare science students was also added.

5.In the methods section, it is mentioned that participatory action research (PAR) was adopted in the research. But, this study does not carry any essence of PAR. Participatory action research requires participants to build and keep evidential records of practice, theory, and reflection and to provide a reasoned justification to others for their work. Doing this requires authentic participation and is collaborative, establishing self-critical, non-hierarchical communities and partnerships. It is also a recursive and systematic process of learning, with planning, action, analysis, and reflection leading to further planning, action, analysis, and reflection. Only, the peer research would be appropriate but it is not action research. The methods section also misses why peer research was used.

I think the reviewer highlights an important point. Thank you for this thoughtful reflection. The research was participatory however doesn’t quite meet the threshold for ‘action’ as you rightly state. It was not an iterative approach in the evidence gathering although it was iterative in skill development. Participatory approach is presented and explored. An extensive description of the peer researcher approach and underpinning rational was added to the methods section. 

6.Ethical issues in doing the qualitative peer research are missing. How these issues were handled?Emphasis was also placed on the ethical challenges. 

7.Conclusion of the study does not hold the essence of the overall result and discussion.

 A review of the manuscript was undertaken, and a revised conclusion is now presented. 

At last, this paper could be made publishable after the rigorous work in the methods section. Does it actually a PAR? Does it complete all the processes of PAR?

 The revision focused on creating a robust methods section, highlighting participatory research and the partnership approach taken within the study. The revision also enhanced the data analysis approach and rigor adopted. 

Thank you for the opportunity to enhance this manuscript. Please do not hesitate to contact me for further reflection or recommendations.

Kind regards

Jean

---

## [Decision Letter · Decision Letter 1]

5 Aug 2022

PONE-D-22-09516R1Participatory peer research exploring the educational impact of Covid-19 on Allied Health and Healthcare Science StudentsPLOS ONE

Dear Dr. Daly-Lynn,

Thank you for submitting your manuscript to PLOS ONE. After careful consideration, we feel that it has merit but does not fully meet PLOS ONE’s publication criteria as it currently stands. Therefore, we invite you to submit a revised version of the manuscript that addresses the points raised during the review process.

If applicable, we recommend that you deposit your laboratory protocols in protocols.io to enhance the reproducibility of your results. Protocols.io assigns your protocol its own identifier (DOI) so that it can be cited independently in the future. For instructions see: https://journals.plos.org/plosone/s/submission-guidelines#loc-laboratory-protocols. Additionally, PLOS ONE offers an option for publishing peer-reviewed Lab Protocol articles, which describe protocols hosted on protocols.io. Read more information on sharing protocols at https://plos.org/protocols?utm_medium=editorial-emailutm_source=authorlettersutm_campaign=protocols.

We look forward to receiving your revised manuscript.

Kind regards,

Gwo-Jen Hwang

Academic Editor

PLOS ONE

Reviewers' comments:

Reviewer's Responses to Questions

**Comments to the Author**

1. If the authors have adequately addressed your comments raised in a previous round of review and you feel that this manuscript is now acceptable for publication, you may indicate that here to bypass the “Comments to the Author” section, enter your conflict of interest statement in the “Confidential to Editor” section, and submit your "Accept" recommendation.

Reviewer #1: (No Response)

Reviewer #2: All comments have been addressed

2. Is the manuscript technically sound, and do the data support the conclusions?

Reviewer #1: Partly

Reviewer #2: Yes

3. Has the statistical analysis been performed appropriately and rigorously? 

Reviewer #1: N/A

Reviewer #2: N/A

4. Have the authors made all data underlying the findings in their manuscript fully available?

Reviewer #1: No

Reviewer #2: Yes

5. Is the manuscript presented in an intelligible fashion and written in standard English?

Reviewer #1: Yes

Reviewer #2: Yes

6. Review Comments to the Author

Reviewer #1: Dear authors,

Thank you for your efforts, they are still some issues; please address them,

1. Due to the title being “Participatory pee 1 r research exploring the educational impact of Covid-19 on Allied Health and Healthcare Science Students” the platform should address how to recruit different professional field participants to join the platform.

Please clarify how to ensure different professional field participants’ baseline, although the authors stated that “All students regularly use this platform to participate in teaching and learning.’”

2. Data analysis section, “The data analysis was undertaken in two separate parts with two different teams. All transcripts were coded by one person and reviewed by two people.” In addition, on page 11, the study presented that “…through data collection and analysis supported the rigorous analytic approach with multiple partners…” please add multiple partners’(coder) reliability and validity.

3. up to the topic is “Qualitative participatory peer research exploring the educational

impact of Covid-19 on Allied Health and Healthcare Science Students” however, the

study related to educational activities on students is unclear, so suggest authors address the study’s educational activities, such as the online class via research activities process, detailed research content, and task. They are unclear.

Thank you.

Reviewer #2: The authors have revised their manuscript as recommended in the first draft. However, I have a remark regarding the role of peer researchers in the process of study. The role of peer researchers is seen only in data collection procedures then how could you say they were peer researchers?

The rest of all is fine and the manuscript can be published.

7. PLOS authors have the option to publish the peer review history of their article (what does this mean?). If published, this will include your full peer review and any attached files.

Reviewer #1: No

Reviewer #2: No

---

## [Author Response · Author response to Decision Letter 1]

14 Sep 2022

Dear Editor and Reviewers,

Thank you for the opportunity to enhance and revise this manuscript. The following letter outlines the authors response to reviewer’s comments.

Reviewer 1 comment Response to reviewer

1. Due to the title being “Participatory pee 1 r research exploring the educational impact of Covid-19 on Allied Health and Healthcare Science Students” the platform should address how to recruit different professional field participants to join the platform.

Please clarify how to ensure different professional field participants’ baseline, although the authors stated that “All students regularly use this platform to participate in teaching and learning.’”

The line in relation to this question was identified on Pg. 9 line 12-13. I believe this query is around the use of the Blackboard Collaborate Ultra platform. Clarifying text has been added line 11 and line 15 to highlight the online meeting room could be accessed by any participant with a meeting link. The entire sample, which is any registered student from the different professional’s fields, would have equal access to participation given that they use Blackboard for their teaching and learning at the University. In addition, the weblink gives access to the meeting room. The participants were recruited from programmes registered within the school of health sciences and therefore had equal opportunity to participate in a focus group.

2. Data analysis section, “The data analysis was undertaken in two separate parts with two different teams. All transcripts were coded by one person and reviewed by two people.” In addition, on page 11, the study presented that “…through data collection and analysis supported the rigorous analytic approach with multiple partners…” please add multiple partners’(coder) reliability and validity. 

Participatory data analysis uses the subjective interpretation of the data by people with lived experience. Through discussion and reflection of this experience richer data collection, discussion and interpretation could be achieved.

Coder reliability and validity was not used as part of this approach and therefore cannot be provided in this manuscript. Coding reliability approaches measure the level of agreement between coders (Guest et al., 2012). The participatory approach uses the DEPICT framework to harness the meaning and understanding of the data by multiple (peer) researchers to create depth not to measure the level of agreement. Pg. 10 line 10-17 seeks to address the reason for the coding book as a scaffolding to support collaborative data analysis. The sentence Pg. 11 line 17-19 stated ‘the consistency of one researcher (JDL) through data collection and analysis supported the rigorous analytic approach with multiple partners.’ A singular researcher enabled consistency and ensured shared meaning across the two teams analysing the data. 

3. up to the topic is “Qualitative participatory peer research exploring the educational

impact of Covid-19 on Allied Health and Healthcare Science Students” however, the

study related to educational activities on students is unclear, so suggest authors address the study’s educational activities, such as the online class via research activities process, detailed research content, and task. They are unclear.

Thank you. Thank you for this comment. 

I have changed the title slightly as a response. This study aims to capture the experiences students had as they navigated the pandemic on their journey to becoming allied health professionals and healthcare scientists. The ‘educational impact’ was misleading as a qualitative piece of work. 

Reviewer 2 comment Response to reviewer

The authors have revised their manuscript as recommended in the first draft. However, I have a remark regarding the role of peer researchers in the process of study. The role of peer researchers is seen only in data collection procedures then how could you say they were peer researchers?

The rest of all is fine and the manuscript can be published.

Thank you for the recommendation to publish this manuscript. Just to highlight the peer researcher role was throughout the lifespan of the project. Pg. 6 line 4-5 sets this out briefly. In addition to collecting the data, peer researchers reviewed the research protocol and ethical documentation, co-designed the topic guide, analysed the data and have supported various different dissemination strands in the project. 

Thank you for the opportunity to enhance this manuscript. Please do not hesitate to contact me for further reflection or recommendations.

Kind regards

Jean

---

## [Decision Letter · Decision Letter 2]

2 Oct 2022

Participatory peer research exploring the experience of learning during Covid-19 for Allied Health and Healthcare Science Students

PONE-D-22-09516R2

Dear Dr. Daly-Lynn,

We’re pleased to inform you that your manuscript has been judged scientifically suitable for publication and will be formally accepted for publication once it meets all outstanding technical requirements.

Kind regards,

Anand Nayyar, Ph.D.

Academic Editor

PLOS ONE

Additional Editor Comments (optional):

The Manuscript stands Accepted with no further revisions.

Reviewers' comments:

Reviewer's Responses to Questions

**Comments to the Author**

1. If the authors have adequately addressed your comments raised in a previous round of review and you feel that this manuscript is now acceptable for publication, you may indicate that here to bypass the “Comments to the Author” section, enter your conflict of interest statement in the “Confidential to Editor” section, and submit your "Accept" recommendation.

Reviewer #1: All comments have been addressed

Reviewer #2: All comments have been addressed

2. Is the manuscript technically sound, and do the data support the conclusions?

Reviewer #1: Yes

Reviewer #2: Yes

3. Has the statistical analysis been performed appropriately and rigorously? 

Reviewer #1: (No Response)

Reviewer #2: N/A

4. Have the authors made all data underlying the findings in their manuscript fully available?

Reviewer #1: Yes

Reviewer #2: Yes

5. Is the manuscript presented in an intelligible fashion and written in standard English?

Reviewer #1: Yes

Reviewer #2: Yes

6. Review Comments to the Author

Reviewer #1: the revised manuscript "Participatory peer research exploring the experience of learning during Covid-19 for Allied Health and Healthcare Science Students " is much better, I have no other new issue.

Thank you.

Reviewer #2: Dear Editor and Authors

I re-read your manuscript and I found it is now in publishable form. My query on the last review was about the role of peer researchers. Authors have responded specifically in the manuscript. I recommend finalizing the manuscript according to the given format and correct grammatical errors if exist. Thank you.

7. PLOS authors have the option to publish the peer review history of their article (what does this mean?). If published, this will include your full peer review and any attached files.

Reviewer #1: No

Reviewer #2: No

---

## [Editor Report · Acceptance letter]

5 Oct 2022

PONE-D-22-09516R2 

Participatory peer research exploring the experience of learning during Covid-19 for Allied Health and Healthcare Science Students 

Dear Dr. Daly-Lynn:

I'm pleased to inform you that your manuscript has been deemed suitable for publication in PLOS ONE. Congratulations! Your manuscript is now with our production department. 

Kind regards, 

on behalf of

Dr. Anand Nayyar 

Academic Editor

PLOS ONE